# Optimizing blood culture diagnostics through laboratory automation: reducing turnaround time and improving clinical outcomes

Juanxiu Qin,[1] Haomin Zhang,[1] Xinyu Zhang,[2] Lihui Wang,[3] Yuetian Yu,[3,4,5] Min Li,[1] Zhen Shen[1]

**ABSTRACT**    Bloodstream infections pose significant clinical challenges, requiring rapid and accurate diagnostics to guide timely antimicrobial therapy. Although efforts to reduce turnaround time (TAT) have focused on emerging molecular techniques and rapid antimicrobial susceptibility testing (AST) methods, these innovations currently complement rather than replace standard protocols. To balance innovation with reliability, we investigated the impact of integrating laboratory automation into the blood culture diagnostic workflow while preserving conventional pathogen identification via mass spectrometry and AST. In this single-center, prospective study, we compared key performance indicators between a historical cohort (March–August 2023; 192 episodes, 110 patients) and an intervention cohort (March–August 2024; 156 episodes, 112 patients) utilizing an automated workflow with the BacT/Alert Virtuo BC system, VPlus 50 automated sample processor, and Vitek 2 XL Microbiology Analyzer. The optimized workflow included automated loading and unloading of blood culture bottles, automatic subculturing of positive cultures, expert system-based review of AST results, and automated reporting of negative results. Post-optimization, the time from blood culture collection to final report was significantly reduced from 95.99 h (79.29–117.63) to 60.81 h (47.06–82.38), with a more pronounced reduction for gram-negative bacteria (44.87 h) compared with gram-positive bacteria (27.24 h). Although the 28-day mortality rate remained unchanged, patients in the post-optimization group experienced significantly shorter hospital stays, faster achievement of optimal therapy, and lower costs related to antibiotics, laboratory tests, and overall hospitalization. These findings indicate that integrating automation in blood culture diagnostics enhances laboratory efficiency, improves clinical outcomes, and reduces healthcare costs, supporting its broader implementation in clinical microbiology.

**IMPORTANCE**    Bloodstream infections require prompt and accurate diagnosis to enable timely, targeted antimicrobial therapy. Traditional blood culture workflows are often hindered by labor-intensive processes and limited after-hours capacity, delaying critical clinical decisions. This study demonstrates that integrating laboratory automation, including automated loading and unloading of blood culture bottles, automatic subculturing of positive cultures, expert system-based review of antimicrobial susceptibility results, and automated reporting of negative results can significantly reduce diagnostic turnaround time without additional staffing. Implemented in a high-volume tertiary hospital, the optimized workflow led to faster initiation of optimal therapy, shorter hospital stays, and reduced healthcare costs, particularly for gram-negative bacteremia. These findings highlight how automation enhances both laboratory efficiency and patient outcomes, supporting its broader adoption in modern clinical microbiology.

**Peer Reviewer** Megan H. Amerson-Brown, The University of Alabama at Birmingham, Birmingham, Alabama, USA

Address correspondence to Zhen Shen, zhenshen@shsmu.edu.cn, or Min Li, rjlimin@shsmu.edu.cn.

Juanxiu Qin and Haomin Zhang contributed equally to this article. Author order was determined both alphabetically and in order of increasing seniority.

The authors declare no conflict of interest.

See the funding table on p. 10.

KEYWORDS    blood culture, laboratory automation, turnaround time, clinical outcome, laboratory workflow

Bloodstream infections are a major healthcare concern, associated with poor clinical outcomes and increasing global incidence. In North America and Europe alone, an estimated 2 million cases occur annually, resulting in approximately 250,000 deaths (1, 2). A 4-year retrospective study evaluating nosocomial bloodstream infections at a tertiary hospital in China reported an incidence of 4.11 per 1,000 admissions and a 28-day mortality rate of 24.4% (3). Timely initiation of appropriate antimicrobial therapy is crucial, as delays in treatment are linked to higher mortality rates (4, 5). Early and precise pathogen identification is essential for optimizing antibiotic therapy and improving patient outcomes (6). Therefore, minimizing the time required for pathogen detection and susceptibility testing remains a key objective in clinical microbiology.

Despite advancements in molecular techniques and biomarkers, blood culture remains the gold standard for diagnosing bloodstream infections (7). However, traditional blood culture methods involve labor-intensive steps, including specimen collection, subculturing, identification (ID), and antimicrobial susceptibility testing (AST). Various strategies have been developed to accelerate laboratory diagnostics (8). Molecular methods can expedite the identification of common pathogens and detect certain resistance genes but cannot replace phenotypic AST (9–11), which remains essential for determining minimal inhibitory concentrations (MICs) and guiding antimicrobial therapy. Emerging rapid AST methods from positive blood cultures have the potential to reduce reporting time, but they cannot be used with polymicrobial blood cultures, and their clinical utility requires further validation (12–14). Ultimately, these novel approaches serve as a complement to the traditional ID and AST techniques and have not been widely implemented in clinical microbiology.

The integration of laboratory automation has significantly enhanced microbial diagnostics, particularly in blood culture processing (15, 16). While preserving traditional ID via matrix-assisted laser desorption ionization-time of flight mass spectrometry (MALDI-TOF MS) and AST, we optimized the blood culture workflow by implementing automated loading and unloading of blood bottles, automatic subculturing of positive cultures, expert system-based review of antimicrobial susceptibility results, and automated reporting of negative results. This streamlined approach has significantly reduced turnaround time (TAT) for blood cultures. To evaluate its effectiveness and clinical impact, we conducted a prospective study comparing key indicators before and after implementation. Our findings revealed that, without increasing labor costs, these workflow improvements substantially reduced TAT, shortened hospital stays, lowered associated costs, and provided notable clinical benefits.

## MATERIALS AND METHODS

### Study design

This study was conducted at Renji Hospital, School of Medicine, Shanghai Jiao Tong University (Shanghai, China), a 1,400-bed university medical center. Blood culture workflow optimization was implemented from March 2024 to August 2024, incorporating automated loading and unloading of blood bottles, automatic subculturing of positive cultures, expert system-based review of antimicrobial susceptibility results, and automated reporting of negative results. These processes were performed using the BacT/Alert Virtuo Blood Culture Automation System (bioMérieux, France), VPlus 50 Fully Automated Microbial Sample Processing System (bioMérieux, France), and Vitek 2 XL Microbiology Analyzer (bioMérieux, France). The optimized blood culture workflow had not been implemented hospital-wide. This study included three departments: Hepatic Surgery Department (HSD), Hepatic Surgery Intensive Care Unit (HSICU), and Surgical Intensive Care Unit (SICU). We selected surgical departments and intensive care units with the highest blood culture volumes and standardized clinical workflows

to ensure consistency and comparability of data. No strict patient matching was performed between the pre- and post-optimization cohorts. Hospitalized patients in the three departments were prospectively monitored following the optimized blood culture workflow. In contrast, historical data from adult patients hospitalized in the same departments between March 2023 and August 2023 were retrospectively reviewed for comparison.

A blood culture set consists of one aerobic bottle and one anaerobic bottle collected at the same time from a single venipuncture site. A bacteremic episode is defined as a single occurrence of bloodstream infection caused by the same bacterial species in a patient. Multiple positive blood cultures from the same patient within 1 week are classified as part of the same episode. Inclusion criteria included being ≥18 years of age and having no known positive blood culture prior to admission. Patients were excluded for discharge, death, or entering hospice care before the final AST. To maintain data accuracy, duplicate blood cultures of the same strain from the same episode were excluded.

## Pre-optimization workflow procedures

The Clinical Microbiology Laboratory operates Monday to Saturday, from 8:00 a.m. to 5:00 p.m. Blood cultures are delivered from the wards within 1 h of collection and are incubated immediately, utilizing satellite incubators during off-hours to ensure continuous 24/7 processing. The laboratory employs the BD BACTEC FX400 system (Becton, Dickinson, USA) for blood culture analysis. Upon arrival, blood culture bottles are manually scanned, entered into the Laboratory Information System (LIS), and then placed into the system for incubation and monitoring. During working hours, Gram staining is performed immediately when a blood culture is flagged as positive, and the results are promptly reported to the hospital. After hours, positive blood cultures remain unprocessed until staff return the next day (Fig. 1). Blood agar plates are inoculated with 0.1 mL samples from positive cultures and incubated at 35°C in a 5% $CO_2$ incubator overnight. The next day, bacterial growth is analyzed using MALDI-TOF MS (Autobio, China) for ID, and the results are reported to clinicians. AST is performed using the Vitek 2 Compact system (bioMérieux, France). On the third day, AST results are submitted to clinicians via the LIS.

## Retrospective multicenter study for blood culture TAT

To provide background on baseline TATs across different hospitals in China, we conducted a retrospective multicenter study across 21 hospitals in five provinces in China during 2021 and 2023. Data extraction included vial codes, collection times, reception times, loading times, and AST reporting times, along with anonymized patient information. TAT for blood cultures was defined as the interval from blood culture collection to the release of the final microbiology report, including organism identification and antimicrobial susceptibility results, to the clinical team. The overall blood culture TATs in Shanghai (74.1–84.3 h), Shandong (73.5–100.9 h), and Hunan (79.0–113.0 h) were consistent with pre-optimization findings in our study (Fig. S1). In contrast, Guangzhou, which had implemented workflow optimizations such as night shifts, achieved shorter TATs (68.8–82.3 h). These results highlight the need for further workflow optimization to shorten TATs and enhance blood culture efficiency in China.

## Optimized blood culture workflow

The VPlus 50 Fully Automated Microbial Sample Processing System (bioMérieux, France) was integrated with the BacT/Alert Virtuo Blood Culture System (bioMérieux, France) to streamline blood culture diagnostics. In this workflow, the Virtuo system handles incubation and positivity detection, whereas the VPlus 50 automates all downstream processing following a positive result. Positive bottles are directly transferred to the VPlus 50 for automated smear preparation, minimizing delays and reducing the risk of contamination associated with manual handling. The VPlus 50 system also performs

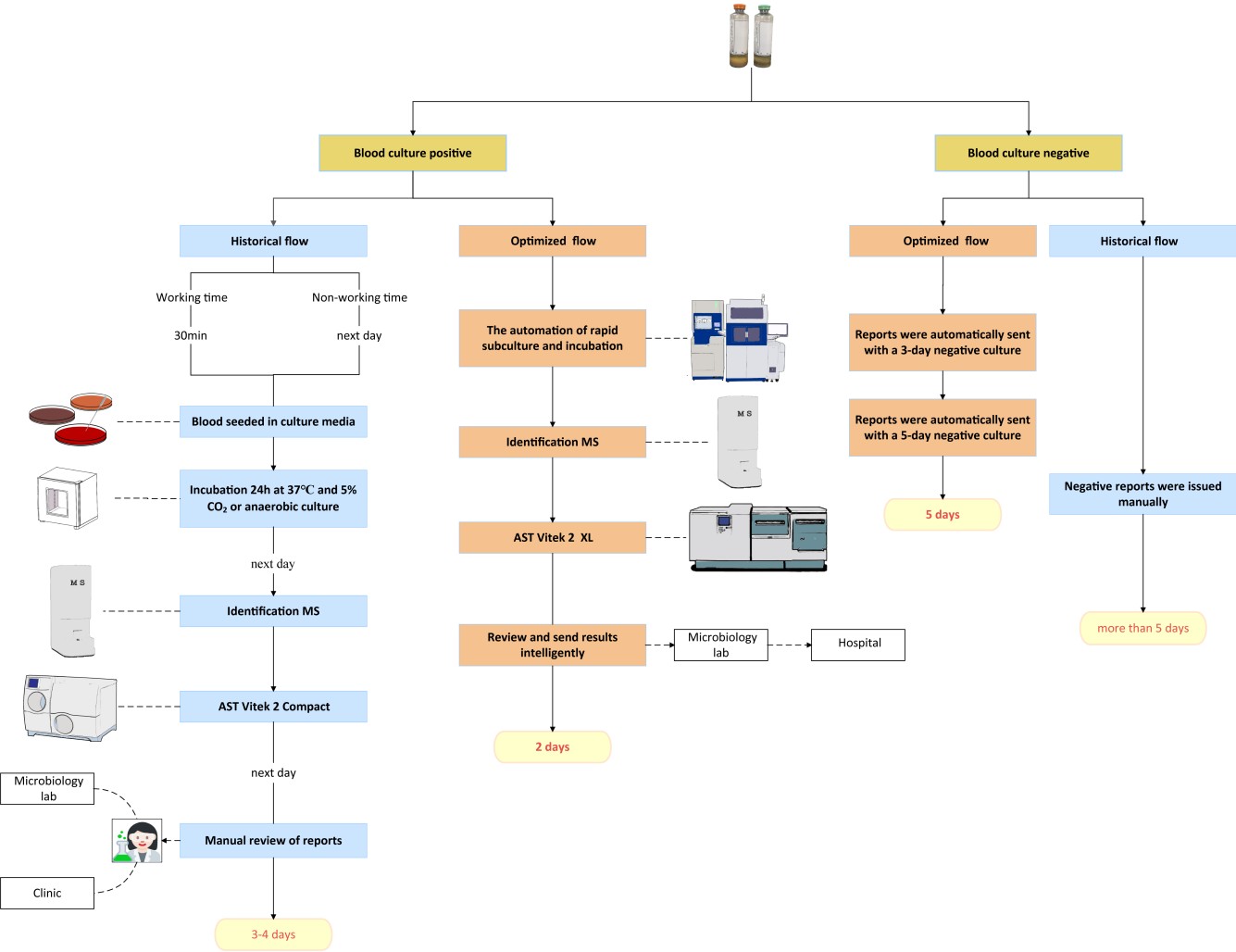

**FIG 1** The historical and optimized blood culture workflows. AST, antimicrobial susceptibility testing; MS, mass spectrometry.

automatic subculturing onto solid media and transfers plates to an integrated incubator, significantly shortening pre-analytical time before pathogen identification (Fig. 1).

Following the subculture of positive blood cultures, a minimum incubation period of 8 h was required for gram-negative bacteria and 12 h for other pathogens to ensure sufficient growth for reliable ID and AST. Subculture strains were rapidly identified via MALDI-TOF MS (Autobio, China), with results promptly communicated to clinical teams for faster treatment decisions. AST was conducted using the Vitek 2 XL Microbiology Analyzer (bioMérieux, France), ensuring efficient and standardized execution. The rule-based expert systems in the Vitek 2 XL enable "automatic review of AST results," and AST results were automatically uploaded to the LIS, eliminating manual data entry and paper-based documentation. To enhance reporting accuracy, a 3-day pre-reporting feature was introduced for negative blood cultures, along with a 5-day automatic review function. Blood cultures submitted outside routine microbiology laboratory working hours were excluded from this analysis. These blood culture bottles were incubated in satellite incubators rather than the Virtuo automated system.

## Data collection and definitions

Patient demographics, comorbidities, specific medications, antibiotic administration data, laboratory data, microbiological data, and clinical data were extracted from the

electronic medical record (EMR). Clinical outcomes, including cure, death, transfer, and automatic discharge, were specifically reported. Microbial contaminants were defined per laboratory guidelines (i.e., 1 of 2 sets positive for an organism rarely pathogenic, such as *Micrococcus*, *Corynebacterium*, or *non-Bacillus anthracis* species) adopted from previously published recommendations (17, 18) or by medical records review. For such contaminants, AST was not performed, and only organism identification was reported. Coagulase-negative staphylococci were routinely reported; however, those determined to be contaminants by clinical evaluation were excluded from the outcome analysis. Achievement of optimal therapy (AOT) was determined retrospectively by infectious disease physicians (18). Optimal therapy was defined as active, narrow-spectrum antibiotics initiated after AST results. Non-optimal therapy included extended broad-spectrum antibiotics or delayed regimens. Inactive therapy refers to treatments with no *in vitro* activity against the isolated pathogen. Time to optimal therapy (TTOT) was the time from blood culture collection to when therapy was determined to be optimal. For example, in a patient with sepsis caused by pan-susceptible *Escherichia coli* initially treated empirically with meropenem, TTOT was measured from culture collection to the point when meropenem was discontinued and a narrower-spectrum agent, such as a piperacillin, a first- to third-generation cephalosporin, or an oral fluoroquinolone, was initiated (18).

In cases involving blood culture contaminants where no antibiotic therapy was deemed necessary, TTOT was considered to be zero to distinguish contaminants, as optimal therapy was defined as withholding treatment. If empiric therapy was determined to be optimal, TTOT was based on the time of empiric therapy initiation or blood culture collection, whichever was later. Antimicrobial costs were counted following blood culture collection. Laboratory outcomes included time from culture collection to Gram stain, ID, and AST. If multiple microorganisms were identified in a single bottle, the reporting times of all microorganisms were included in TAT calculations.

## Statistical analysis

Statistical analysis was performed using Stata version 18 (College Station, USA). Normality of continuous data was assessed using the Shapiro-Wilk test. As most variables did not follow a normal distribution, continuous variables were reported as medians with interquartile ranges (IQRs), whereas categorical variables were expressed as frequencies and percentages. Differences between the post-optimization and pre-optimization cohorts were assessed using the Mann-Whitney U test for continuous variables and the chi-squared test for categorical variables. Statistical significance was defined as a *P*-value < 0.05.

## RESULTS

### Baseline demographics and clinical characteristics

A total of 222 eligible patients were included in this study, with 110 in the pre-optimization group and 112 in the post-optimization group (Table 1). The median age was 59 years, and 72.1% were male (Table 1). ICU admissions accounted for 59.0% of cases. The most common sources of infection were the gastrointestinal system (34.2%) and respiratory system (8.1%), whereas 46.4% of patients had an unidentified source. There were no significant differences between the two groups in terms of age, gender, ICU admission, APACHE II score, infection source, and related laboratory findings.

### Impact of laboratory automation on blood culture turnaround time

To assess the effect of laboratory process optimization, we analyzed the time from blood culture collection to Gram stain reporting and the final report. The time from blood culture collection to final report was significantly reduced post-optimization, from 95.99 h (79.29–117.63) to 60.81 h (47.06–82.38) (*P* < 0.01) (Table 2). The time to Gram stain

**TABLE 1** Baseline patient demographics and clinical characteristics[a]

| Characteristics | Total (n = 222) | Pre-optimization (n = 110) | Post-optimization (n = 112) | P value |
|---|---|---|---|---|
| Age | 59 [45, 70] | 60 [46, 70] | 58.0 [44.25, 60] | 0.42 |
| Sex | | | | 0.40 |
| Male | 160 (72.1) | 82 (74.5) | 78 (69.6) | |
| Female | 62 (27.9) | 28 (25.5) | 34 (30.4) | |
| Wards | | | | 0.19 |
| ICUs | 131 (59.0) | 61 (55.5) | 70 (62.5) | |
| N-ICUs | 91 (41.0) | 49 (44.5) | 42 (37.5) | |
| Comorbidities | | | | 0.28 |
| <3 | 121 (54.5) | 64 (58.2) | 57 (50.9) | |
| ≥3 | 101 (45.5) | 46 (41.8) | 55 (49.1) | |
| Liver transplantation | 54 (24.3) | 25 (22.7) | 29 (25.9) | 0.58 |
| APACHE II score | 20.65 [15.03, 24.81] | 20.35 [15.26, 24.58] | 21.07 [14.86, 25.21] | 0.42 |
| Source of bacteremia | | | | |
| Respiratory tract | 18 (8.1) | 9 (8.2) | 9 (8.0) | 0.96 |
| Urinary tract | 9 (4.1) | 4 (3.6) | 5 (4.5) | 0.75 |
| Digestive system | 76 (34.2) | 33 (30.2) | 43 (38.4) | 0.19 |
| Catheter-related | 9 (4.1) | 5 (4.5) | 4 (3.6) | 0.71 |
| Donor-derived infections | 6 (2.7) | 5 (4.5) | 1 (0.9) | 0.12 |
| Reproductive system | 1 (0.5) | 1 (0.9) | 0 | 0.50 |
| Unknown | 103 (46.4) | 53 (48.2) | 50 (44.6) | 0.60 |
| Laboratory testing | | | | |
| WBC count (×10$^9$ /L) | 9.84 [6.47, 14.50] | 10.41 [6.75, 15.45] | 9.42 [6.35, 13.81] | 0.21 |
| Neutrophil count (×10$^9$ /L) | 8.72 [5.11, 12.43] | 8.99 [5.55, 13.12] | 8.05 [4.88, 11.29] | 0.15 |
| CRP (mg/L) | 63.01 [27.50, 125.65] | 73.99 [30.65, 117.34] | 48.38 [22.94, 124.73] | 0.81 |
| PCT (ng/ml) | 1.90 [0.41, 9.90] | 1.84 [0.43, 11.85] | 1.93 [0.32, 7.66] | 0.46 |

[a]Data are presented as median [interquartile range] or number (percentage). ICU, intensive care unit; APACHE, Acute Physiology and Chronic Health Evaluation; WBC, white blood cell; CRP, C-reactive protein; PCT, procalcitonin.

reporting was also approximately 1 h faster in the post-optimization group, although the difference was not statistically significant (P = 0.26).

## Bloodstream infection pathogens and turnaround times

During the study period, a total of 3,265 blood culture sets were collected. Of these, seven sets in the historical cohort and nine sets in the optimized cohort were identified as contaminants. Notably, a higher proportion of patients in the optimized cohort had two blood culture sets collected. The blood culture positivity rate increased from 12.5% (208/1,658) in the pre-optimization phase to 17.2% (277/1,607) post-optimization (Fig. 2). The higher positivity rate observed in the post-optimization cohort can be attributed to several factors: First, the FN Plus blood culture bottles used with the BacT/Alert Virtuo system offer enhanced antibiotic adsorption, improving microbial recovery, particularly in patients receiving antimicrobial therapy. Second, the closed-system incubator maintains stable incubation temperatures (35–37°C), reducing thermal fluctuations that could otherwise inhibit microbial growth and delay detection. To accurately assess TATs, multiple positive blood cultures from the same patient within 1 week are classified as part of the same episode, resulting in the analysis of 348 unique episodes and bacterial isolates. Among these, 49.1% (171/348) were gram-positive and 37.6% (131/348) were gram-negative bacteria (Table 2). The remaining 46 isolates consisted of anaerobes (n = 9), yeasts (n = 26), and polymicrobial cultures (n = 11).

The reduction of the time from blood culture collection to final report was more pronounced for gram-negative bacteria than for gram-positive bacteria (44.87 h vs. 27.24 h decrease). Among gram-positive bacteria, the greatest TAT reduction was observed in *Enterococcus* (92.60 vs. 59.15 h), followed by coagulase-negative *Staphylococcus* (94.42 vs. 68.66 h). *Streptococcus* spp. also showed a decrease (82.89 vs. 69.76 h), although without

**TABLE 2** Bloodstream infection pathogens and turnaround time (TAT)[a]

| | No. (%) | Pre-optimization group | | Post-optimization group | | P value |
|---|---|---|---|---|---|---|
| | | n (%) | TAT | n (%) | TAT | |
| Positive blood culture episodes | 348 | 192 | 95.99 [79.29, 117.63] | 156 | 60.81 [47.06, 82.38] | **<0.01** |
| Gram stain | 348 | 192 | 28.18 [23.24, 51.72] | 156 | 27.02 [22.85, 40.46] | 0.26 |
| Gram-positive bacteria | 171 (49.1) | 94 (49.0) | 91.37 [76.87, 109.14] | 77 (49.4) | 64.13 [53.84, 82.01] | **<0.01** |
| *Staphylococcus aureus* | 4 (1.1) | 1 (0.5) | 82.95 | 3 (1.9) | 60.11 [56.35, 95.35] | /[b] |
| CoNS | 102 (29.3) | 56 (29.2) | 94.42 [78.34, 108.98] | 46 (29.5) | 68.66 [55.59, 83.65] | **<0.01** |
| *Enterococcus* spp. | 42 (12.1) | 21 (10.9) | 92.60 [80.75, 119.02] | 21 (13.5) | 59.15 [41.37, 69.65] | **<0.01** |
| *Streptococcus* spp. | 7 (2.0) | 3 (1.6) | 82.89 [73.04, 84.90] | 4 (2.6) | 69.76 [65.00, 82.80] | 0.72 |
| Others | 16 (4.6) | 13 (6.8) | 80.27 [64.73, 107.98] | 3 (1.9) | 92.98 [80.06, 120.01] | 0.46 |
| Gram-negative bacteria | 131 (37.6) | 72 (37.5) | 95.64 [75.12, 117.21] | 59 (37.8) | 50.77 [39.51, 65.37] | **0.01** |
| *Escherichia coli* | 30 (8.6) | 15 (7.8) | 103.95 [85.68, 126.86] | 15 (9.6) | 47.97 [42.46, 55.19] | **0.01** |
| *Klebsiella* spp. | 37 (10.6) | 18 (9.4) | 100.40 [74.72, 140.28] | 19 (12.2) | 46.44 [36.24, 54.78] | **0.01** |
| *Enterobacter* spp. | 16 (4.6) | 7 (3.6) | 108.34 [92.15, 121.88] | 9 (5.8) | 54.94 [34.86, 73.64] | **0.04** |
| *Pseudomonas* spp. | 18 (5.2) | 13 (6.8) | 91.65 [73.14, 97.03] | 5 (3.2) | 61.61 [39.08, 79.39] | 0.18 |
| *Acinetobacter* spp | 23 (6.6) | 15 (7.8) | 90.39 [68.40, 120.57] | 8 (5.1) | 42.04 [36.82, 52.07] | **0.01** |
| Others | 7 (2.0) | 4 (2.1) | 103.19 [89.30, 134.27] | 3 (1.9) | 78.46 [71.29, 81.79] | **0.03** |
| Anaerobic bacteria | 9 (2.9) | 4 (2.6) | 135.99 [110.06, 163.41] | 5 (3.2) | 161.52 [87.23, 164.93] | 1.00 |
| Yeast | 26 (7.5) | 17 (8.9) | 113.93 [97.82, 135.06] | 9 (5.8) | 108.34 [92.17, 167.84] | 0.94 |
| *Candida* | 24 (6.9) | 15 (7.8) | 113.93 [99.43, 137.21] | 9 (5.8) | 108.34 [92.17, 167.84] | 0.93 |
| *Cryptococcus neoformans* | 2 (0.6) | 2 (1.0) | 112.91 [103.53, 122.28] | 0 (0) | / | / |
| Polymicrobial | 11 (3.2) | 5 (2.6) | 87.02 [86.76, 140.32] | 6 (3.8) | 67.93 [54.10, 81.68] | **0.03** |
| Negative blood culture episodes | 1319 | 703 | 135.03 [125.95, 141.45] | 616 | 127.04 [122.64, 134.59] | **0.01** |

[a]Data are presented as median [interquartile range] or number (percentage). Statistically significant values are highlighted in bold (P < 0.05). CoNS, coagulase-negative staphylococci; TAT, turnaround time, from blood culture collection to the release of the final microbiology report.
[b]/ indicates not applicable.

statistical significance. Fungal TATs also decreased (113.93 vs. 108.34 h), although the difference was not statistically significant (P = 0.94).

For gram-negative bacteria, significant reductions were observed in *Escherichia coli* (103.95 vs. 47.97 h), *Klebsiella* (100.4 vs. 46.44 h), *Enterobacter* (108.34 vs. 54.94 h), and *Acinetobacter* (90.39 vs. 42.04 h). Although *Pseudomonas aeruginosa* (91.65 vs. 61.61 h) also exhibited reductions, it was not statistically significant (Table 2). In contrast, TATs for gram-positive bacilli and anaerobic bacteria increased (80.27 vs. 92.98 h and 135.99 vs. 161.52 h, respectively), likely due to limitations in the automated system's anaerobic culture capabilities. Additionally, the automated review function reduced the reporting time for negative blood cultures by nearly 8 h (135.03 vs. 127.04 h).

## Impact of optimization on clinical outcomes and hospital costs

As shown in Table 3, there was no significant difference in 28-day mortality rates between the pre- and post-optimization groups (15.5% vs. 14.3%, P = 0.81). However, the total hospital stay was significantly shorter in the post-optimization group compared with the pre-optimization group (19 days vs. 29 days, P = 0.04). Optimal therapy was achieved more frequently in the post-optimization group (92.0%) compared with the pre-optimization group (81.8%) (P = 0.02), as well as more quickly (median TTOT, 42.4 h) compared with the pre-optimization group (60.6 h) (P = 0.02) (Table 3). Additionally, costs related to antibiotics, laboratory tests, and total hospitalization expenses were significantly reduced (269.16 vs. 678.59, 1,637.15 vs. 2,417.38, and 14,847.92 vs. 24,334.96, respectively, with P < 0.05 for all comparisons). These findings suggest that workflow optimization not only reduced the financial burden associated with empiric antimicrobial therapy but also provided measurable clinical benefits for patients.

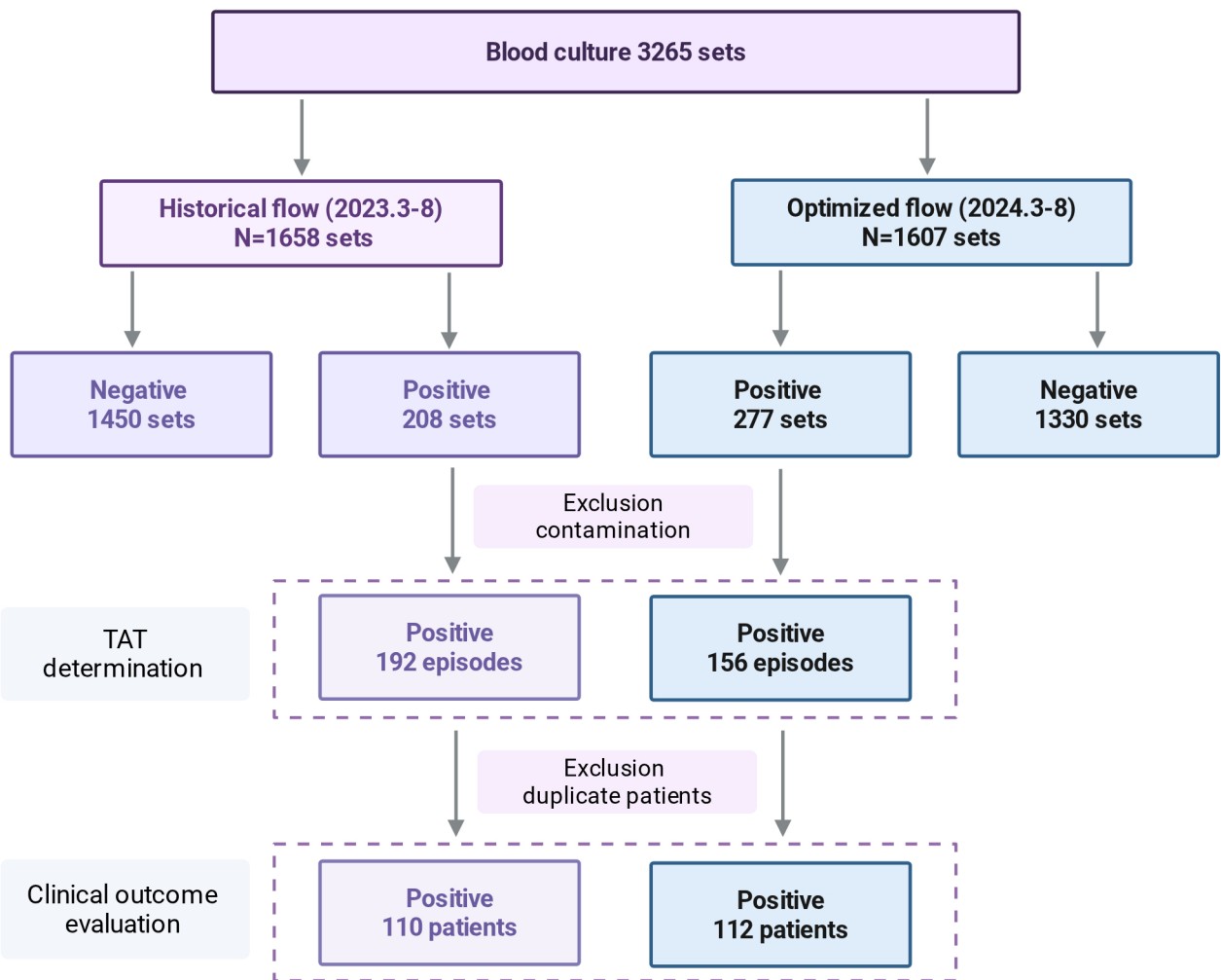

**FIG 2** The flow chart depicts the workflow and participants included during the study periods. A blood culture set consists of one aerobic bottle and one anaerobic bottle collected at the same time from a single venipuncture site. A bacteremic episode is defined as a single occurrence of bloodstream infection caused by the same bacterial species in a patient. Multiple positive blood cultures from the same patient within 1 week are classified as part of the same episode. A total of seven blood culture sets in the historical workflow and nine in the optimized workflow were identified as contaminants. TAT, turnaround time.

## DISCUSSION

Decreasing the time for bloodstream infection diagnosis is paramount for reducing its impact on mortality and healthcare costs. As a result, there has been growing interest in enhancing the speed and accuracy of blood culture diagnostics in clinical microbiology laboratories. Efforts to reduce TAT have largely focused on emerging molecular detection techniques and rapid AST methods (7, 8). However, these novel approaches currently serve as adjuncts rather than replacements for standard diagnostic protocols. To balance innovation with reliability, we optimized the blood culture workflow while preserving traditional organism ID via mass spectrometry and AST. Key improvements included automatic subculturing of positive cultures, expert system-based review of AST results, and automated reporting of negative results. This streamlined approach significantly reduced TAT, leading to shorter hospital stays, lower healthcare costs, and improved clinical outcomes, all without increasing labor costs.

The integration of laboratory automation has established a new standard for bloodstream infection diagnostics, transforming laboratory workflows from an intermittent to a continuous schedule. Our retrospective multicenter study revealed that over 50% of "flag positive" blood cultures occur outside regular working hours;

**TABLE 3** Impact of blood culture workflow optimization on clinical outcomes and hospital costs[a]

| Outcome | Total (n = 222) | Pre-optimization group (n = 110) | Post-optimization group (n = 112) | P value |
|---|---|---|---|---|
| Clinical outcome | | | | 0.90 |
| Cure | 166 (74.8) | 82 (74.5) | 84 (75) | |
| Death | 33 (14.9) | 17 (15.5) | 16 (14.3) | |
| Transfer | 3 (1.3) | 2 (1.8) | 1 (0.9) | |
| Automatic discharge | 20 (9) | 9 (8.2) | 11 (9.8) | |
| 28-day mortality | 33 (14.9) | 17 (15.5) | 16 (14.3) | 0.81 |
| Optimal therapy | | | | |
| Achieved optimal therapy | 193 (86.9) | 90 (81.8) | 103 (92.0) | **0.02** |
| TTOT | 52.5 [6.1, 84.3] | 60.6 [7.5, 91.2] | 42.4 [4.8, 78.9] | **0.02** |
| Hospital stays (days) | 20 [11, 33] | 28 [11, 38] | 19 [10, 30] | **0.04** |
| Total hospital costs | 18,807.08 [8,656.67, 31,809.19] | 24,334.96 [12,183.30, 35,954.98] | 14,847.92 [7,294.45, 28,578.84] | **0.01** |
| Laboratory test costs | 1,945.94 [1,045.18, 3,734.45] | 2,417.38 [1,079.45, 4,107.22] | 1,637.15 [1,005.94, 3,096.08] | **0.04** |
| Antimicrobial costs | 422.63 [114.65, 1,474.09] | 678.59 [153.93, 2,813.65] | 269.16 [87.87, 966.41] | **<0.01** |

[a]Data are presented as median [interquartile range] or number (percentage). Statistically significant values are highlighted in bold (P < 0.05). Healthcare costs were calculated in dollars. Cost data were extracted from hospital billing records and reflect the total expenses incurred during the entire hospitalization associated with each bacteremic episode. TTOT, time to optimal therapy following blood culture collection.

however, most laboratories do not process or report them until the next day, a challenge previously noted in multiple studies (19–21). Although extending laboratory services to a 24/7 schedule could reorganize workflows and reduce TAT, economic and labor constraints often make this impractical (7). In contrast, our study demonstrated that automated processing of positive blood cultures reduced TAT by 35 h, achieving efficiency gains without additional procedures beyond standard methods.

The reduction in time from blood culture collection to the final report was more significant for gram-negative bacteria than for gram-positive bacteria (44.87 vs. 27.24 h). Approximately 50% of all positive cultures were collected between 8:00 a.m. and 11:00 a.m. Due to their faster growth, gram-negative bacteria, which generally grow faster, typically become positive within 8–16 h and often flag positive overnight when collected earlier in the day. With automated subculturing, identification, and antimicrobial susceptibility testing can be conducted the following day, significantly reducing turnaround time. In contrast, gram-positive bacteria have a longer time-to-positivity and require at least 12 h of incubation after automated subculture before further processing. Similarly, fungi such as *Candida* species exhibit even slower growth kinetics, which inherently delays both positivity and downstream processing. Although automation improves reporting time for gram-positive bacteremia, the reduction is less pronounced compared to gram-negative cases.

Before implementing laboratory automation in the blood culture workflow, our clinical microbiology laboratory had developed rapid identification and antimicrobial resistance gene detection methods for positive blood cultures, guided by previous studies. MALDI-TOF was applied directly to positive blood culture bottles for rapid pathogen identification (22, 23), whereas Xpert Carba-R was used to detect carbapenemase genes (24). These tests were conducted only after clinician consultation following Gram stain results and were not part of the standard blood culture protocol. The results were communicated to the clinical team with a clear disclaimer that they were for reference only, with final ID and AST results serving as the definitive diagnostic criteria. To prevent potential bias in therapeutic decisions, blood cultures processed using these rapid methods were excluded from the analysis.

The optimized workflow did not result in a statistically significant difference in 28-day mortality rates between the pre- and post-optimization groups. One potential explanation for the lack of mortality difference may be that mortality in bloodstream infections is multifactorial, influenced not only by the timeliness of microbiological diagnosis but also by patient comorbidities, severity of illness, and the effectiveness of subsequent clinical management (6). Although reducing TAT is critical for prompt and appropriate antibiotic administration, its impact on mortality might be mitigated by these confounding factors.

In other words, earlier laboratory results can facilitate faster therapeutic adjustments; however, the overall outcome may still depend on the patient's underlying condition and the presence of complications that are beyond the scope of laboratory efficiency improvements.

This study has several limitations. First, as this was a single-center study conducted primarily among surgical patients, the results may not be fully generalizable to other institutions with different laboratory workflows, patient populations, or healthcare infrastructures. However, our findings provide a strong foundation for future multicenter studies to validate the impact of laboratory automation in diverse settings. Second, the quasi-experimental design lacked patient matching, and additional factors may have influenced clinical outcomes. Although randomized controlled trials could provide stronger causal evidence, our real-world data demonstrate the practical benefits of automation in a high-volume clinical microbiology laboratory. Finally, we did not assess the long-term impact of automation on laboratory workload and staff efficiency. Future studies should explore how automation affects laboratory personnel, cost-effectiveness, and workflow sustainability over extended periods.

In conclusion, our study demonstrates that laboratory automation and workflow optimization significantly enhance blood culture processing efficiency, shorten hospital stays, and reduce healthcare costs without additional staffing requirements. These findings support broader implementation of automated systems in clinical microbiology to optimize patient outcomes and healthcare resource utilization.

## ACKNOWLEDGMENTS

This study was supported by the National Natural Science Foundation of China (82272374), and the 3‑Year Action Plan for Strengthening Public Health System in Shanghai (GWVI‑11.1‑11). The funders had no role in study design, data collection and analysis, decision to publish, or preparation of the manuscript.

## AUTHOR AFFILIATIONS

[1]Department of Laboratory Medicine, Renji Hospital, School of Medicine, Shanghai Jiao Tong University, Shanghai, China
[2]Department of Medical Administration, Renji Hospital, School of Medicine, Shanghai Jiao Tong University, Shanghai, China
[3]Department of Critical Care Medicine, Renji Hospital, School of Medicine, Shanghai Jiaotong University, Shanghai, China
[4]Key Laboratory of Multiple Organ Failure (Zhejiang University), Ministry of Education, Zhejiang, China
[5]Key Laboratory of Intelligent Pharmacy and Individualized Therapy of Huzhou, Huzhou, China

## AUTHOR ORCIDs

Lihui Wang  http://orcid.org/0000-0002-8339-8848
Yuetian Yu  http://orcid.org/0000-0002-0193-4046
Min Li  http://orcid.org/0000-0002-8571-2909
Zhen Shen  http://orcid.org/0009-0006-1034-0705

## FUNDING

| Funder | Grant(s) | Author(s) |
| --- | --- | --- |
| National Natural Science Foundation of China | 82272374 | Zhen Shen |

## AUTHOR CONTRIBUTIONS

Juanxiu Qin, Data curation, Formal analysis, Investigation, Methodology, Software, Writing – original draft | Haomin Zhang, Data curation, Formal analysis, Investigation,

Methodology, Resources, Software, Writing – original draft | Xinyu Zhang, Data curation, Formal analysis, Investigation, Software | Lihui Wang, Data curation, Formal analysis, Investigation, Methodology, Software | Yuetian Yu, Conceptualization, Investigation, Methodology, Project administration, Resources, Supervision | Min Li, Conceptualization, Project administration, Supervision, Writing – review and editing | Zhen Shen, Conceptualization, Data curation, Funding acquisition, Methodology, Project administration, Supervision, Writing – review and editing

## ETHICS APPROVAL

This study was approved by the Ethics Committee of Renji Hospital, School of Medicine, Shanghai Jiao Tong University (Reference No. LY2024-296-C). Sample collection and result reporting were conducted as part of standard patient care. All data were de-identified prior to access, and no personal information was stored in the study database.

## ADDITIONAL FILES

The following material is available online.

### Supplemental Material

**Fig. S1 (Spectrum01927-25-s0001.docx).** Turnaround times for blood cultures across different regions in mainland China.

### Open Peer Review

**PEER REVIEW HISTORY (review-history.pdf).** An accounting of the reviewer comments and feedback.

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
