## [Reviewer comments · Microbiology Spectrum]

Microbiology Spectrum

Optimizing Blood Culture Diagnostics Through Laboratory Automation: Reducing Turnaround Time and Improving Clinical Outcomes

Juanxiu Qin, Haomin Zhang, Xinyu Zhang, Lihui Wang, Yuetian Yu, Min Li, and Zhen Shen

Corresponding Author(s): Zhen Shen, Shanghai Jiao Tong University School of Medicine Affiliated Renji Hospital Department of Laboratory Medicine

Review Timeline:

Submission Date:	July 8, 2025
Editorial Decision:	August 18, 2025
Revision Received:	September 3, 2025
Editorial Decision:	September 23, 2025
Revision Received:	October 3, 2025
Accepted:	October 21, 2025

Editor: Vera Tesic

Reviewer(s): Disclosure of reviewer identity is with reference to reviewer comments included in decision letter(s). The following individuals involved in review of your submission have agreed to reveal their identity: Megan H Amerson-Brown (Reviewer #1)

Transaction Report:

DOI: <https://doi.org/10.1128/spectrum.01927-25>

Re: Spectrum01927-25 (**Optimizing Blood Culture Diagnostics Through Laboratory Automation: Reducing Turnaround Time and Improving Clinical Outcomes**)

Dear Dr. Zhen Shen:

Thank you for the privilege of reviewing your work. Below you will find my comments, instructions from the Spectrum editorial office, and the reviewer comments.

Revision Guidelines

Sincerely,
Vera Tesic
Editor
Microbiology Spectrum

Reviewer #1 (Comments for the Author):

The authors describe a study in which laboratory automation was used to improve patient care and costs for bacteremia. The results of this study are consistent and support similar findings associated with blood culture identification panels and rapid phenotypic susceptibility testing platforms (basically any platform or workflow that would allow for more information and better results to be transmitted to the clinician at an earlier time).

Line62-63. It is unclear why US and European epidemiology data is used to support a study evaluating outcomes at a single hospital in China. Please use data related to the location of your study to support the importance of this manuscript.

Throughout document the reference to "intelligent susceptibility review" is confusing? If this refers to the Vitek interpretation software for AST results this is standard on the vitek 2 instrument and is not new technology. Intelligent susceptibility review gives the perception that AI is being used when it is not.

If this system was automated how did the automation know which bottles were from these specific units and should go on the automation platform versus bottles from other units in the hospital.

How were second bottles handled from "possible contaminants" that were part of the historical workflow group handled? Were these excluded? The exclusion criteria throughout the study is very unclear especially regarding extensive non-working hours of the lab? I would expect that all cultures that are positive during non-working hours would be excluded since the "standard of care" would not be available during this time; however, this does not seem to be the case or is written in a manner that clarifies this.

Table 1. Why is there only one p value for comparison between the two groups for large subgroups? Why was the comparison not made for each individual characteristic?

It is unclear if contaminated blood cultures were included or excluded as part of this study. In one portion of the manuscript, it is mentioned that they were excluded, however they are given a TTOT score in a later section. Please clarify the role of blood cultures deemed to be contaminated in this study.

The study mentions that optimization reduces costs. The cost of the instrument should be addressed so laboratorians understand the cost of capital compared to cost reduction per patient. There is clearly a cost to the lab for bringing in new equipment.

Why does the figure show an AI symbol going from microbiology lab to the clinic?

Since the study was performed in surgical units surgery should be listed on the characteristics tab and evaluated for statistical difference between the groups. APACHE score or a score indicating that there was not significant difference in patient severity should also be included.

If the only thing that is being optimized in this study is automation of the blood culture from the blood culture instrument to plating wouldn't a better analysis be time to positivity of growth on agar plates? The other factors involved in testing would not be impacted by this automation

The discussion should include a hypothesis as to why the impact of this automation had such a great impact on Gram-negative bacteremia but did not have as great of an impact on fungal and Gram-positive bacteremia.

Response to Reviewers

We thank the reviewer for the detailed and constructive comments, which have greatly helped us improve the quality and clarity of our manuscript. Below we provide a point-by-point response to each comment. Revisions made in the manuscript are highlighted for clarity.

Reviewer #1 (Comments for the Author):

The authors describe a study in which laboratory automation was used to improve patient care and costs for bacteremia. The results of this study are consistent and support similar findings associated with blood culture identification panels and rapid phenotypic susceptibility testing platforms (basically any platform or workflow that would allow for more information and better results to be transmitted to the clinician at an earlier time).

Response: We thank the reviewer for the positive feedback and acknowledgment of our study.

Line 62-63. It is unclear why US and European epidemiology data is used to support a study evaluating outcomes at a single hospital in China. Please use data related to the location of your study to support the importance of this manuscript.

Response: We thank the reviewer for this valuable comment. We have revised the Introduction to include epidemiological data on bloodstream infections in China to better support the relevance of our study (Lines 63-65).

Throughout document the reference to "intelligent susceptibility review" is confusing? If this refers to the Vitek interpretation software for AST results this is standard on the vitek 2

instrument and is not new technology. Intelligent susceptibility review gives the perception that AI is being used when it is not.

Response: We thank the reviewer for this helpful clarification. As noted, the term “intelligent susceptibility review” refers to the Vitek interpretation software for AST results, which functions as a rule-based expert system rather than AI. We have replaced this term with “expert system-based review of antimicrobial susceptibility results” throughout the manuscript to avoid any potential misunderstanding (Lines 35, 53-54, 86-87, and 98-99).

If this system was automated how did the automation know which bottles were from these specific units and should go on the automation platform versus bottles from other units in the hospital.

Response: The automated loading function eliminated the need for manual barcode scanning, as the blood culture system automatically read and placed the bottles. For this study, only bottles from the three designated surgical departments were loaded onto the BacT/Alert Virtuo Blood Culture Automation System (Lines 102-105).

How were second bottles handled from "possible contaminants" that were part of the historical workflow group handled? Were these excluded? The exclusion criteria throughout the study is very unclear especially regarding extensive non-working hours of the lab? I would expect that all cultures that are positive during non-working hours would be excluded since the "standard of care" would not be available during this time; however, this does not seem to be the case or is written in a manner that clarifies this.

Response: Microbial contaminants were defined according to laboratory guidelines (e.g., growth of organisms rarely considered pathogenic, such as *Micrococcus*, *Corynebacterium*, or non-anthraxis *Bacillus* species in only one of two sets), consistent with published recommendations and verified through medical record review. All contaminants were excluded from the analysis, regardless of whether cultures were collected during or outside routine working hours. The main distinction between the historical and optimized workflows lies in the handling of positive cultures during non-working hours. In the optimized workflow, the VPlus 50 Fully Automated Microbial Sample Processing System was integrated with the BacT/Alert Virtuo, enabling automatic subculturing onto solid media and incubation, thereby significantly reducing pre-analytical delays. This clarification has been added to the manuscript (Lines 145-153, 169-175).

Table 1. Why is there only one p value for comparison between the two groups for large subgroups? Why was the comparison not made for each individual characteristic?

Response: We have revised Table 1 to include p-values for each individual characteristic in the comparison between the two groups.

It is unclear if contaminated blood cultures were included or excluded as part of this study. In one portion of the manuscript, it is mentioned that they were excluded, however they are given a TTOT score in a later section. Please clarify the role of blood cultures deemed to be contaminated in this study.

Response: We thank the reviewer for this comment. Contaminated blood cultures were excluded from the study. TTOT was assigned a value of zero only to distinguish contaminants, and we have clarified this point in the revised manuscript (Lines 185-187).

The study mentions that optimization reduces costs. The cost of the instrument should be addressed so laboratorians understand the cost of capital compared to cost reduction per patient. There is clearly a cost to the lab for bringing in new equipment.

Response: We thank the reviewer for this important comment. Our analysis focused on costs related to antibiotics, laboratory testing, and overall hospitalization. Instrument acquisition costs were not included, as another blood culture instrument is also in place for the pre-optimization cohort.

Why does the figure show an AI symbol going from microbiology lab to the clinic?

Response: The AI symbol in Figure 1 has been deleted.

Since the study was performed in surgical units surgery should be listed on the characteristics tab and evaluated for statistical difference between the groups. APACHE score or a score indicating that there was not significant difference in patient severity should also be included.

Response: Surgery and APACHE scores have been added to Table 1 and evaluated for statistical differences between the groups. No significant differences were observed between the pre-optimization and post-optimization cohorts (Lines 212-214).

If the only thing that is being optimized in this study is automation of the blood culture from the blood culture instrument to plating wouldn't a better analysis be time to positivity of growth on agar plates? The other factors involved in testing would not be impacted by this automation.

Response: We thank the reviewer for this comment. We have evaluated the time required for visible growth on agar plates. After subculture of positive blood cultures, a minimum of 8 hours was required for Gram-negative bacteria and 12 hours for other pathogens to achieve sufficient growth for reliable identification and AST. This clarification has been added to the manuscript (Lines 154-156).

The discussion should include a hypothesis as to why the impact of this automation had such a great impact on Gram-negative bacteremia but did not have as great of an impact on fungal and Gram-positive bacteremia.

Response: We have added a hypothesis in the Discussion to explain the differential impact of automation. Approximately 50% of all positive cultures were collected between 8:00 a.m. and 11:00 a.m. Due to their faster growth, Gram-negative bacteria typically flagged positive overnight when collected the same day. With automated subculturing, identification and susceptibility testing could then be performed the next day, substantially shortening turnaround time. In contrast, Gram-positive bacteria generally require longer incubation before positivity and at least 12 additional hours after automated subculture, limiting the extent of time reduction. Similarly, fungi such as *Candida* species exhibit even slower growth kinetics, which inherently delays both positivity and downstream processing. As a result, while automation improved efficiency across all organism groups, its impact was most pronounced for Gram-negative bacteremia (Lines 286-296).

Re: Spectrum01927-25R1 (**Optimizing Blood Culture Diagnostics Through Laboratory Automation: Reducing Turnaround Time and Improving Clinical Outcomes**)

Dear Dr. Zhen Shen:

Thank you for the privilege of reviewing your work. Below you will find my comments, instructions from the Spectrum editorial office, and the reviewer comments.

Revision Guidelines

Sincerely,
Vera Tesic
Editor
Microbiology Spectrum

Reviewer #2 (Comments for the Author):

Reviewer Work

Recommend using matrix absorption laser desorption ionization-time of flight mass spectrometry (MALDI-TOF MS) for all instances making reference to mass spectrometry in the text and figures

Based on the primary end point (28-day mortality) not showing differences before and after implementation of the automated workflow, I do not think the paper supports the claim of improvement in clinical outcomes. It looks like the automated workflow manifested in shorter hospital stays, faster implementation of adequate antibiotic therapy (which remains poorly defined, see below), and lower costs, but the authors did not provide a justification how these changes improved clinical outcomes (it is likely the workflow change has some clinical benefits, perhaps not seen in a single-center study, or based on other secondary end points that the authors did not evaluate, like adverse reactions attributed to the prolonged use of broad-spectrum antibiotic therapy (like bug-drug mismatches, readmission rates, hospital-onset *Clostridioides difficile* infection, decreased adverse and/or allergic reactions to antimicrobials, lesser incidence of HAI across the board, to name a potential secondary clinical end points).

It is interesting to see some of the particularities of laboratory operation in other countries when reviewing papers, like the availability of microbiology laboratory staff to process samples, and I cannot help to think about the laboratory workflow I am lucky to enjoy in my hospital system. I agree that phenotypical testing is the gold standard for determining the best course of antimicrobial therapy. However, a lot can be known by rapid molecular testing, at least in my experience. For example, the authors mention the use of rapid testing to detect carbapenemase activity (line 305). Implementation of Verigene for Gram positive and Gram negative organisms without clinician intervention in our facility allows rapid changes in antimicrobial therapy pending standard susceptibility testing, and in most occasions, it helps narrow down the antimicrobial spectrum of activity. Removing these instances in the final analysis (line 309-310), as you made clear in the discussion section (which should have been mentioned in the Methods section) likely creates bias in analyzing turnaround times (TAT).

Line 80-81. "...and have not been widely performed in ...". Not sure if what you are trying to convey here is the lack of studies in this area (perhaps "...and have not been widely implemented in..." ?

Line 87-88. Based on prior studies, or your experience? If from prior studies, references should be added. If a result of this study, this should go in the Conclusion section. See discussion regarding clinical benefits above.

Line 107-110. The study design seems to suggest all patients matching inclusion criteria were included in the analysis. Table 1 suggests both groups were similar (which also weakens the argument of why there were no changes in the primary end point, as claimed on Lines 311-315). The authors mention the lack of patient matching as a weakness (line 325-326). I think the lack of patient matching should be mentioned in the Method section too.

Lines 115-116. "No known positive blood cultures" and "no previous bacteremia during admission" are somewhat confusing statements. It either seems to be the same, or I wonder whether the second statement is to state other episodes of bacteremia during the admission.

Line 127. "Clinic" should be "hospital" given you are performing this study in the hospital setting (same issue is in the Figure 1 workflow)

Lines 135-145. This paragraph is somewhat strange. It seems to be a reference to some previous work in improving TAT across hospitals in different facilities across some provinces, but there is no mention about the sort of changes (except for the changes implemented in Guangzhou, lines 142-143). Without the specifics of these changes, I do not think the authors can claim these findings support the results of this study (line 142). I would be much more interested to see this work published! I think this paragraph belongs to the Background section, without any mentions about how this supports the results of the study (especially when you have not even mentioned the methods for the background work, or this work in the article).

Lines 165-168. Could you provide a rationale or reason for this decision?

Line 174. Should be "non-Bacillus anthracis species"

Line 177. Who performed the clinical evaluation? The surgical teams or a dedicated infectious diseases/pharmacy stewardship program or consultation?

Line 180. "overly broad" seems rather vague. The paper does not delve too much into antimicrobial stewardship activities in your facility. Suppose you have a patient admitted with ascending cholangitis to one of your surgical wards, and the patient is started on broad-spectrum antibiotic therapy pending culture data, based on clinical guidelines and local microbiology data. It might be broad, but it does not make it wrong necessarily.

Line 186. "penicillin" seems to be a bad example here. Perhaps an aminopenicillin or piperacillin is what you want to use here. Some people would argue that using a third-generation or quinolone therapy in a patient with *E. coli* bacteremia susceptible to ampicillin or cephalexin is equal to keep using a broad-spectrum agent. It all depends of the scenario.

Line 197. Add (College Station, USA) after Stata version 18

Lines 214-215. I would prefer "gastrointestinal" to "digestive". You can drop the word "infection" after "system", "respiratory", and

"unidentified".

Lines 228-229. Was there a reason or a hypothesis for this? Automated systems might require some changes in preanalytical protocols (assuring proper collection for the automated system to perform the expected way?).

Line 243-244. Result about fungal isolates should be presented later (not sure why it ended up sandwiched in this paragraph). It should follow the organization you have on your Table 2, perhaps after the paragraph ending on line 256.

Line 252. I did not see any distinction of Gram positive bacilli on Table 2, so I am wondering why this needs to be entered here and compared to anaerobic bacteria. I am assuming that when you talk about Gram positive bacilli you are considering things like *Listeria*, *Corynebacterium*, or *Bacillus*, distinct from anaerobic Gram positive bacilli like *Clostridium*, for example. I think this needs to be added to Table 2 if you are planning to mention it in the text. There was no mention of statistical analysis for these groups either.

Line 280. The study failed to prove a statistically significant change in clinical outcome based on your primary end point, so this is not supported by your data. Regarding the labor costs, the study did not show any information about the cost of implementing the automated workflow (training and/or hiring of personnel, price of instruments and supplies, information technology requirements to connect the laboratory information system to the electronic health record and viceversa), which has to be contrasted with the savings mentioned by the authors.

Lines 287. Would this be true for all facilities? I wonder if this might not be practical or cost effective for larger institutions.

Line 292. Any hypotheses for this?

Line 298. *Candida* in italics

Line 315. True but at least based on your data, this does not seem to be the case (both groups were similar)

Line 322. Or even to other areas within your facility, which might have different patient's profiles in regards to age, comorbidities, and disease severity indexes.

Line 334. I think this is the first instance you mentioned the automated workflow did not create additional staffing requirements (see comment about line 280).

Table 3. "Clinical outcomes" should have been mentioned in the Methods section as secondary end points.

Figure 2. I wonder if this figure can be rearranged or presented in a different manner. Given that reducing TAT is one of the goals here, a time-based figure might be more useful, perhaps similar to a Gantt chart with time on the X-axis and presenting the workflow steps within time frames to highlight the differences in regard to time. I think this gets lost in the way the figure is designed. The illustrations are an artistic license I suppose.

Response to Reviewers

We thank the reviewer for the detailed and constructive comments, which have greatly helped us improve the quality and clarity of our manuscript. Below we provide a point-by-point response to each comment. Revisions made in the manuscript are highlighted for clarity.

Reviewer #2 (Comments for the Author):

Recommend using matrix absorption laser desorption ionization-time of flight mass spectrometry (MALDI-TOF MS) for all instances making reference to mass spectrometry in the text and figures.

Response: We thank the reviewer for the suggestion. All references to mass spectrometry in the text and figures have been updated to “matrix-assisted laser desorption/ionization–time of flight mass spectrometry (MALDI-TOF MS).”

Based on the primary end point (28-day mortality) not showing differences before and after implementation of the automated workflow, I do not think the paper supports the claim of improvement in clinical outcomes. It looks like the automated workflow manifested in shorter hospital stays, faster implementation of adequate antibiotic therapy (which remains poorly defined, see below), and lower costs, but the authors did not provide a justification how these changes improved clinical outcomes (it is

likely the workflow change has some clinical benefits, perhaps not seen in a single-center study, or based on other secondary end points that the authors did not evaluate, like adverse reactions attributed to the prolonged use of broad-spectrum antibiotic therapy (like bug-drug mismatches, readmission rates, hospital-onset Clostridioides difficile infection, decreased adverse and/or allergic reactions to antimicrobials, lesser incidence of HAI across the board, to name a potential secondary clinical end points).

Response: We thank the reviewer for this thoughtful comment. The optimized workflow did not result in a statistically significant difference in 28-day mortality between the pre- and post-optimization groups. Mortality in bloodstream infections is multifactorial, influenced not only by the timeliness of microbiological diagnosis but also by patient comorbidities, severity of illness, and subsequent clinical management. While reduced turnaround time facilitates earlier adjustments to antimicrobial therapy, the overall effect on mortality may be limited by these confounding factors. This consideration has been discussed in the revised Discussion section (Lines: 309-318).

It is interesting to see some of the particularities of laboratory operation in other countries when reviewing papers, like the availability of microbiology laboratory staff to process samples, and I cannot help to think about the laboratory workflow I am lucky to enjoy in my hospital system. I agree that phenotypical testing is the gold standard for determining the best course of antimicrobial therapy. However, a lot can be known by rapid molecular testing, at least in my experience. For example, the

authors mention the use of rapid testing to detect carbapenemase activity (line 305). Implementation of Verigene for Gram positive and Gram negative organisms without clinician intervention in our facility allows rapid changes in antimicrobial therapy pending standard susceptibility testing, and in most occasions, it helps narrow down the antimicrobial spectrum of activity. Removing these instances in the final analysis (line 309-310), as you made clear in the discussion section likely creates bias in analyzing turnaround times (TAT).

Response: Rapid molecular testing was not part of the routine workflow and was applied only in selected clinical scenarios at the discretion of clinicians. As only a small proportion of patients underwent rapid molecular testing or direct MALDI-TOF identification from positive blood culture bottles, these cases were excluded from the final analysis to minimize bias and maintain consistency in interpreting turnaround time (TAT) (Lines: 301-308).

Line 80-81. "...and have not been widely performed in ...". Not sure if what you are trying to convey here is the lack of studies in this area (perhaps "...and have not been widely implemented in..." ?

Response: Thank you for the suggestion. The sentence has been revised to: *"Ultimately, these novel approaches serve as a complement to traditional ID and AST techniques and have not been widely implemented in clinical microbiology."* (Lines: 79-81)

Line 87-88. Based on prior studies, or your experience? If from prior studies, references should be added. If a result of this study, this should go in the Conclusion section. See discussion regarding clinical benefits above.

Response: This statement summarizes our study results, which demonstrated a significant reduction in blood culture turnaround time (TAT). We have ensured that this finding is discussed in the Discussion section and highlighted appropriately in the Conclusion (Lines: 274-277, 330-332).

Line 107-110. The study design seems to suggest all patients matching inclusion criteria were included in the analysis. Table 1 suggests both groups were similar (which also weakens the argument of why there were no changes in the primary end point, as claimed on Lines 311-315). The authors mention the lack of patient matching as a weakness (line 325-326). I think the lack of patient matching should be mentioned in the Method section too.

Response: We did not perform strict patient matching. However, baseline demographics and clinical characteristics were comparable between the pre- and post-optimization groups, supporting their similarity. We have also noted the lack of patient matching as a methodological limitation in the Methods section, as suggested (Lines: 107-108).

Lines 115-116. "No known positive blood cultures" and "no previous bacteremia during admission" are somewhat confusing statements. It either seems to be the same, or I wonder whether the second statement is to state other episodes of bacteremia during the admission.

Response: Thank you for pointing out this ambiguity. We have revised the inclusion criteria to read: *"No known positive blood cultures prior to admission."* This integrates the two phrases and provides clearer wording (Lines: 115-116).

Line 127. "Clinic" should be "hospital" given you are performing this study in the hospital setting (same issue is in the Figure 1 workflow)

Response: The term "clinic" has been replaced with "hospital", including in the Figure 1 workflow (Lines: 126-127).

Lines 135-145. This paragraph is somewhat strange. It seems to be a reference to some previous work in improving TAT across hospitals in different facilities across some provinces, but there is no mention about the sort of changes (except for the changes implemented in Guangzhou, lines 142-143). Without the specifics of these changes, I do not think the authors can claim these findings support the results of this study (line 142). I would be much more interested to see this work published! I think this paragraph belongs to the Background section, without any mentions about how this supports the results of the study (especially when you have not even mentioned

the methods for the background work, or this work in the article).

Response: Thank you for this helpful comment. The retrospective multicenter study was performed to provide background on baseline TATs across different hospitals in China. The overall blood culture TATs in Shanghai (74.1-84.3 hours), Shandong (73.5-100.9 hours), and Hunan (79.0-113.0 hours) were consistent with our pre-optimization findings, while Guangzhou, which had implemented workflow optimizations such as night shifts, achieved shorter TATs (68.8-82.3 hours). We have clarified in the revised manuscript that this information is presented as background context to illustrate baseline performance, without implying that it directly supports the results of our study (Lines: 134-136, 140-144).

Lines 165-168. Could you provide a rationale or reason for this decision?

Response: Microbial contaminants were defined per laboratory guidelines (ie, 1 of 2 sets positive for an organism rarely pathogenic such as Micrococcus, Corynebacterium, or Bacillus species non-anthraxis) adopted from previously published recommendations (References 17 and 18).

Line 174. Should be "non-Bacillus anthracis species"

Response: Changed accordingly (Lines: 172-173).

Line 177. Who performed the clinical evaluation? The surgical teams or a dedicated infectious diseases/pharmacy stewardship program or consultation?

Response: Achievement of optimal therapy (AOT) was determined retrospectively by dedicated infectious disease physicians (Lines: 177-178).

Line 180. "overly broad" seems rather vague. The paper does not delve too much into antimicrobial stewardship activities in your facility. Suppose you have a patient admitted with ascending cholangitis to one of your surgical wards, and the patient is started on broad-spectrum antibiotic therapy pending culture data, based on clinical guidelines and local microbiology data. It might be broad, but it does not make it wrong necessarily.

Response: We have revised the wording from “overly broad” to “extended broad-spectrum antibiotics” to provide greater clarity and precision (Lines: 179-180).

Line 186. "penicillin" seems to be a bad example here. Perhaps an aminopenicillin or piperacillin is what you want to use here. Some people would argue that using a third-generation or quinolone therapy in a patient with E.coli bacteremia susceptible to ampicillin or cephalexin is equal to keep using a broad-spectrum agent. It all depends of the scenario.

Response: We thank the reviewer for the suggestion. As recommended, “penicillin” has been replaced with “piperacillin” in the manuscript to provide a more appropriate

example (Lines: 182-186).

Line 197. Add (College Station, USA) after Stata version 18

Response: We have added “College Station, USA” after Stata version 18 as suggested (Line: 196).

Lines 214-215. I would prefer "gastrointestinal" to "digestive". You can drop the word "infection" after "system", "respiratory", and "unidentified".

Response: We have revised the terminology as suggested, replacing “digestive” with “gastrointestinal” and removing the word “infection” after “system,” “respiratory,” and “unidentified” (Lines: 212-214).

Lines 228-229. Was there a reason or a hypothesis for this? Automated systems might require some changes in preanalytical protocols (assuring proper collection for the automated system to perform the expected way?).

Response: The higher positivity rate observed in the post-optimization cohort is likely due to multiple factors. First, the FN Plus blood culture bottles used with the BacT/Alert Virtuo system provide enhanced antibiotic adsorption, which improves microbial recovery, particularly in patients receiving antimicrobial therapy. Second, the closed-system incubator maintains stable temperatures (35-37°C), minimizing

thermal fluctuations that could otherwise inhibit microbial growth and delay detection (Lines: 228-233).

Line 243-244. Result about fungal isolates should be presented later (not sure why it ended up sandwiched in this paragraph). It should follow the organization you have on your Table 2, perhaps after the paragraph ending on line 256.

Response: We thank the reviewer for the suggestion. The results regarding fungal isolates have been repositioned to follow the organization in Table 2 and are now presented at the end of the relevant paragraph (Lines: 244-245).

Line 252. I did not see any distinction of Gram positive bacilli on Table 2, so I am wondering why this needs to be entered here and compared to anaerobic bacteria. I am assuming that when you talk about Gram positive bacilli you are considering things like Listeria, Corynebacterium, or Bacillus, distinct from anaerobic Gram positive bacilli like Clostridium, for example. I think this needs to be added to Table 2 if you are planning to mention it in the text. There was no mention of statistical analysis for these groups either.

Response: We thank the reviewer for the comment. We did not perform a direct comparison of TAT between Gram-positive bacilli and anaerobic bacteria. We only aimed to report that the TATs increased for both groups: Gram-positive bacilli (80.27 vs. 92.98 hours) and anaerobic bacteria (135.99 vs. 161.52 hours) (Lines: 249-252).

Line 280. The study failed to prove a statistically significant change in clinical outcome based on your primary end point, so this is not supported by your data. Regarding the labor costs, the study did not show any information about the cost of implementing the automated workflow (training and/or hiring of personnel, price of instruments and supplies, information technology requirements to connect the laboratory information system to the electronic health record and viceversa), which has to be contrasted with the savings mentioned by the authors.

Response: We thank the reviewer for this important comment. Our analysis focused on costs related to antibiotics, laboratory testing, and overall hospitalization. Instrument acquisition costs were not included, as another blood culture instrument is also in place for the pre-optimization cohort.

Lines 287. Would this be true for all facilities? I wonder if this might not be practical or cost effective for larger institutions.

Response: We thank the reviewer for the comment. While some teaching hospitals in Shanghai have implemented 24/7 laboratory services and reorganized workflows to reduce TAT, economic and labor constraints often make this approach impractical for many institutions (Lines: 282-284).

Line 292. Any hypotheses for this?

Response: Approximately 50% of all positive cultures were collected between 8:00 a.m. and 11:00 a.m. Gram-negative bacteria, which generally grow faster, typically become positive within 8-16 hours and often flag positive overnight when collected earlier in the day (Lines: 289-291).

Line 298. Candida in italics

Response: Changed accordingly (Line: 295).

Line 315. True but at least based on your data, this does not seem to be the case (both groups were similar)

Response: We appreciate the reviewer's comment. Mortality in bloodstream infections is multifactorial, influenced not only by the timeliness of microbiological diagnosis but also by patient comorbidities, illness severity, and the effectiveness of subsequent clinical management. Although reducing TAT is important for prompt and appropriate antibiotic therapy, its effect on mortality may be attenuated by these confounding factors (Lines: 309-318).

Line 334. I think this is the first instance you mentioned the automated workflow did not create additional staffing requirements (see comment about line 280).

Response: The manuscript clarifies in the Importance, Introduction, and Discussion

sections that the automated workflow did not require additional staffing (Lines: 51-55, 90-92, 274-277, and 330-332)

Table 3. "Clinical outcomes" should have been mentioned in the Methods section as secondary end points.

Response: We thank the reviewer for the comment. Clinical outcomes-including cure, death, transfer, and automatic discharge-were described in the Methods section as secondary endpoints (Lines: 170-171).

Figure 2. I wonder if this figure can be rearranged or presented in a different manner. Given that reducing TAT is one of the goals here, a time-based figure might be more useful, perhaps similar to a Gantt chart with time on the X-axis and presenting the workflow steps within time frames to highlight the differences in regard to time. I think this gets lost in the way the figure is designed. The illustrations are an artistic license I suppose.

Response: We thank the reviewer for the suggestion. Figure 2 illustrates the study workflow and is not intended to depict TAT. The reduction in TAT for each group, along with statistical significance, is presented in Table 2.

Re: Spectrum01927-25R2 (**Optimizing Blood Culture Diagnostics Through Laboratory Automation: Reducing Turnaround Time and Improving Clinical Outcomes**)

Dear Dr. Zhen Shen:

Your manuscript has been accepted, and I am forwarding it to the ASM production staff for publication. Your paper will first be checked to make sure all elements meet the technical requirements. ASM staff will contact you if anything needs to be revised before copyediting and production can begin. Otherwise, you will be notified when your proofs are ready to be viewed.

Sincerely,
Vera Tesic
Editor
Microbiology Spectrum

Reviewer #2 (Comments for the Author):

Thank you for addressing some of our concerns about the methodology and analysis of the data obtained from this retrospective study.